# The Role of the Maridi Dam in Causing an Onchocerciasis-Associated Epilepsy Epidemic in Maridi, South Sudan: An Epidemiological, Sociological, and Entomological Study

**DOI:** 10.3390/pathogens9040315

**Published:** 2020-04-24

**Authors:** T. L. Lakwo, S. Raimon, M. Tionga, J. N. Siewe Fodjo, P. Alinda, W. J. Sebit, J. Y. Carter, R. Colebunders

**Affiliations:** 1Vector Control Division, Ministry of Health, P.O. Box 1661 Kampala, Uganda; tlakwo@gmail.com (T.L.L.); papeteralinda@gmail.com (P.A.); 2Amref Health Africa, P.O. 30125 Juba, South Sudan; Stephen.Jada@amref.org (S.R.); Moses.Tionga@amref.org (M.T.); 3Global Health Institute, University of Antwerp, 2610 Antwerp, Belgium; josephnelson.siewefodjo@uantwerpen.be; 4National Public Health Laboratory, Ministry of Health, May Rd, P.O. 30125 Juba, South Sudan; wilson.sebit@hotmail.com; 5Amref Health Africa Headquarters, P.O. Box 27691−00506 Nairobi, Kenya; Jane.Carter@Amref.org

**Keywords:** onchocerciasis, epilepsy, nodding syndrome, dam, vector control, elimination, blackflies

## Abstract

**Background**: An epilepsy prevalence of 4.4% was documented in onchocerciasis-endemic villages close to the Maridi River in South Sudan. We investigated the role of the Maridi dam in causing an onchocerciasis-associated epilepsy epidemic in these villages. **Methods**: Affected communities were visited in November 2019 to conduct focus group discussions with village elders and assess the OV16 seroprevalence in 3- to 9-year-old children. Entomological assessments to map blackfly breeding sites and determine biting rates around the Maridi River were conducted. Historical data regarding various activities at the Maridi dam were obtained from the administrative authorities. **Results**: The Maridi dam was constructed in 1954–1955. Village elders reported an increasing number of children developing epilepsy, including nodding syndrome, from the early 1990s. Kazana 2 (the village closest to the dam; epilepsy prevalence 11.9%) had the highest OV16 seroprevalence: 40.0% among children 3–6 years old and 66.7% among children 7–9 years old. The Maridi dam spillway was found to be the only Simulium damnosum breeding site along the river, with biting rates reaching 202 flies/man/h. **Conclusion**: Onchocerciasis transmission rates are high in Maridi. Suitable breeding conditions at the Maridi dam, coupled with suboptimal onchocerciasis control measures, have probably played a major role in causing an epilepsy (including nodding syndrome) epidemic in the Maridi area.

## 1. Introduction

Onchocerciasis (river blindness) is one of the recognized neglected tropical diseases (NTD) of mankind. The disease is caused by the filarial parasite *Onchocerca volvulus* transmitted by blackflies of the genus *Simulium* that breed in fast flowing rivers and streams [1]. Female blackflies are responsible for the transfer of the parasite from one person to another in the process of feeding [1]. Clinical presentations of onchocerciasis include skin and eye diseases as well as various forms of epilepsy and developmental defects in children, thus contributing to its severe public health impact [2].

Efforts to combat onchocerciasis through mass drug administration (MDA) with ivermectin since the early 1990s have registered great success in the Americas [3] and some parts of Africa [4]. These successes have led to a paradigm shift from control to elimination of onchocerciasis [4,5]. However, models and field assessments indicate that ivermectin treatment alone is probably not sufficient to interrupt transmission in regions of Africa where vector densities are high; in such areas, it is necessary to supplement MDA with other interventions [6]. Experiences from Uganda have demonstrated that a combination of vector control and MDA can accelerate the interruption of transmission, thus supporting the important contribution of vector control [7].

Onchocerciasis is a disease of public health importance in Maridi County, South Sudan, and is mostly associated with epilepsy and nodding syndrome [8,9]. Indeed, communities living close to the Maridi dam often complain of an intense blackfly nuisance and frequent seizure disorders. In 2018, in a door-to-door survey, an epilepsy prevalence of about 4.4% and annual incidence of 373.9/100,000 PY was documented in onchocerciasis-endemic villages close to the Maridi River; 85.2% of persons with epilepsy met the criteria of onchocerciasis-associated epilepsy (OAE) [8,9]. No pigs are kept in the Maridi area excluding neurocysticercosis as a cause of this high epilepsy prevalence. The highest prevalence (11.9%) was observed in Kazana 2, a village close to the Maridi dam [8]. For many years, MDA with ivermectin was interrupted in the area; only 40.8% of the population took ivermectin in 2017 [8]. In this study we investigated the role of the Maridi dam in causing an epilepsy epidemic including nodding syndrome in the Maridi area.

## 2. Materials and Methods

### 2.1. Description of the Study Area

Maridi County is situated in the Western Equatoria region of South Sudan. The population is estimated to be 101,065 [10]. This county is endemic for onchocerciasis and faces a high burden of disease caused by epilepsy including nodding syndrome [8,9]. The area has been targeted for MDA with ivermectin by the national NTD Control Programme with the support of partners. The main river is the Maridi River that flows northwards and is fast flowing with several shallow rapids providing suitable breeding sites for blackflies. The Maridi dam (N:4°53′41″; E: 29°27′27.5″) along the Maridi River was built in 1955 to provide water to Maridi town and this dam was repaired in early 2000, according to information from the South Sudan Urban National Water Corporation. The dam has a spillway measuring 132 m across with a metallic foot bridge that connects Kazana 1 and 2 villages. Fast flowing water from the dam overflow is always observed at the spillway and provides a conducive environment for blackfly breeding (Figure 1).

The main streams joining the Maridi River are the Itri and Mbalala Rivers on the Yei road, and the Marindu, Mabulindi, Molisikanga and Manguo Rivers. Other major rivers in the region include the Bahr Naam River in the east which flows through Mvolo and has earlier been reported to be responsible for onchocerciasis transmission in that area [11,12]. In the immediate west is the Ibba River but the status of this river, as far as onchocerciasis transmission is concerned, is unknown; this also applies to the Tonj River to the west. All these rivers join before entering into the Bahr-el-Ghazal River near Bentiu in the northern part of the country. The communities along the Maridi River are engaged in subsistence farming owing to the very fertile soil in the area. The main crops grown are cassava, potatoes, millet, sesame and coffee. Those living close to the Maridi dam do small scale fishing. The population has generally been disturbed by the chronic conflict in the country and resettlements are still ongoing in some of the villages surveyed. The villages most affected by the blackfly biting nuisance are Kazana 1, Kazana 2 and Matara; all these are within a 5 km distance from the Maridi dam. A map showing the sites surveyed and mapped on the Maridi River and its tributaries is shown in Figure 2.

### 2.2. Focus Group Discussions and in Depth Interviews

Focus group discussions (FGDs were organized among the community concerning knowledge and perceptions about onchocerciasis and epilepsy. In-depth interviews were carried out with elders of the village to investigate whether there had been a change in epilepsy incidence including nodding syndrome over the years. The research team piloted the interview and FGD guides to ensure they were well understood by the study participants, and final topic guides were adapted. Arrangements were made to identify suitable locations for interviews and FGD that allowed for confidentiality. All participants provided written informed consent. In-depth interviews were conducted by one researcher while for the FGD researchers worked in pairs with one serving as the moderator and the other as the note taker.

### 2.3. OV16 Testing

A total of 144 children between the ages of 3 and 9 years were tested for onchocerciasis antibodies using the OV16 rapid test (Standard diagnostics, Inc., Gyeonggi-do, Republic of Korea) as an indicator of the degree of recent *O. volvulus* transmission. Blood was obtained by finger prick and the test was performed according to the manufacturer’s instructions. Among these OV16-tested children, we investigated ivermectin use in 2019 for those aged 5–9 years.

### 2.4. Entomological Study

An entomological assessment was carried out to map the breeding sites of blackflies (*Simulium damnosum, s.l*.) on the Maridi River and its tributaries. Investigations were carried out from 26 November to 11 December 2019 at the beginning of the dry season. Fifteen men, aged 20–35 years, were trained on basic blackfly control techniques which included human landing catches (HLC), fly preservation and data collection during the catches. A practical session for the trainees was conducted at the Maridi dam site.

#### 2.4.1. Search for Breeding Sites on the Maridi River

Streams and rivers in central Maridi were visited to collect aquatic stages of the *Simulium* species. Sites were selected based on the map provided since the team did not have an appropriate 1:50,000 map. Usually 30–45 min were spent at each site examining submerged leaves, grasses, sticks and stones for larvae and pupae; rock surfaces were also inspected. Any movable substrates with larvae or pupae attached were placed in a plastic bag, labelled with the site identification, placed in cool box and transported to the Maridi Health Sciences Institute. The mapping of the Maridi River followed conventional methods including the marking of all the breeding sites of *S. damnosum* [13].

#### 2.4.2. Establishment of Catching Sites along the Maridi River

Based on the information on productive vector breeding sites, three catching sites were established at the Maridi dam (20 m from the spillway); Kazana 2 (600 m from the dam) and Matara (3.5 km downstream on the Maridi River). These catching sites were all located in first-line villages (<5 km from river), and the distance between catching sites ranged from between 600 m and 3.5 km since only three communities along the Maridi River experienced intense blackfly biting.

#### 2.4.3. Human Landing Catches (HLC) of Adult Biting *Simulium* along the Maridi River

The collection of biting *Simulium* flies to provide baseline data on biting rates was conducted for seven consecutive days. The catchers at each of the three sites (A, B, C) sat on a chair or log with their legs bare below the knees. Any flies landing on the legs were caught by inverting a small plastic tube over them. After capture some of the flies were kept alive by wrapping them in a moist hand towel wetted in water. This 11 h catch was intended to determine the hourly biting activity and total number of flies that would bite a stationary person between 7:00 and 18:00 h. The two vector collectors worked on alternate hours at established sites on the Maridi River according to HLC standard procedures [14,15].

#### 2.4.4. Dissection of Flies for Age Group Determination (Parity)

Adult biting *S. damnosum* caught by HLC from catching sites A, B, and C were kept alive by wrapping them in a moist towel wetted in water. These flies were transported to the Maridi Health Sciences Institute where a room was assigned as a temporary laboratory. The flies were killed by ether vapor that was substituted for chloroform [16] before putting them on a slide which was transferred to a stereo-microscope. The species was confirmed before the process of opening the abdomen began. Dissection for parity was conducted following standard routine procedures [17,18] and flies were recorded as nulliparous and parous on a dissection form.

### 2.5. Data Analysis

The collected data were entered into spreadsheets and analyzed using R version 3.6.2 (R Core Team, Vienna, Austria). Continuous variables were summarized as means and compared using the Mann–Whitney U test, while categorical variables were expressed as percentages and compared using Chi-squared tests. P-values below 0.05 were considered statistically significant. For the qualitative data, the researchers cross-checked the responses for bias, transcribed them and translated them into English. The English translation was reviewed again by the social scientist to ensure comprehensibility for persons without specific local knowledge. Themes were identified through careful reading and re-reading of the script, and if patterns were recognized, these emerging themes became the categories for analysis.

### 2.6. Ethical Considerations

The protocol for this study was approved by the ethics committees of the Ministry of Health of South Sudan and the University of Antwerp (Ethics Approval Reference number: B300201940004). All volunteering individuals who provided signed informed consent were included in the study. Children younger than 18 years provided their assent to participate under the witness of a consenting adult parent or guardian. All data were codified and treated confidentially.

## 3. Results

### 3.1. Focus Group Discussions (FGD)

Five FGD (groups of 8–10 persons) with a total of 47 participants (27 males and 20 females) were conducted with community leaders in five villages: Kazana 1, Kazana 2, Matara, Gabat and Tarawa in Maridi County. The discussions explored the following main themes: the burden of onchocerciasis and onchocerciasis-associated morbidities, the use of ivermectin in the community, how the blackflies affect the lives of people living in these villages, the local methods employed by the community to combat the blackflies, the views of the community concerning the slash and clear intervention to be introduced to destroy the breeding habitat of the blackflies, and their willingness to support this intervention.

During the FGD, most of participants recognized Kazana as the area with the highest burden of onchocerciasis-associated epilepsy and nodding syndrome with intense blackfly biting.

“The one that is badly affected is an area called Kazana, sometimes we reach there and you will see that two, three or four children in one home are affected”. A 39-year-old man in Hai Tarawa-Maridi.

“They are all over in Maridi here. Most of them are here in Kazana 2”. A 36-year-old man in Kazana 2, Maridi.

“They are many in Kazana (dam) if they bite you, you will scratch yourself until tomorrow, it is itching badly. If it is now itching like this, it is a way of OV signs showing. The blood is poisoned and the body is now affected”. A 55-year-old man in Kazana 1, Maridi.

“During the time of war, when we returned home from the bush, we found the dam was so dirty, these small flies started to increase there like bees, then they started biting people seriously, then people started asking from where do these flies come? They said from Kazana (dam), then the name Kazana flies appeared, from there we realized an increase in number of children with this disease (nodding), but we don’t know if it is the cause or not”. A 52-year-old woman, Kazana 1 village.

The participants also discussed the way blackflies breed. They had mixed views about where exactly they breed and the best way they can be controlled. Others expressed misconceptions about the blackflies’ habitat and breeding sites. One participant had a comment on the slash and clear strategy.

“These flies live in holes, they breed there and the method you said to remove them (slash and clear) cannot kill them, because they live in holes and these holes are many here, they hide themselves during the dry season in it. I believe they live in places with quiet water, and holes, and the best way to kill them is by applying medicines to the holes and water”. A 41-year-old man, Matara village.

It was suggested that with the building of the dam, blackflies started to appear.

“The dam was built in 1955, there was no flies by then, even during the time of Arab (the period after the independence of Sudan) they were not there, then after the war (Sudan civil war in 1983) the flies started to come, previously there was no flies and the area was a road people were moving through it and it was clean no dirt or grass, then at the time of SPLA (when the revolution movement (SPLA) took control of the town) in what year I don’t know the flies started to come and breed, they found their ground to stay comfortably, Chinese have come recently and they found these flies were biting people seriously, they saw that is bad, they poured medicines there and this relieved people from these flies for two years”. An elderly man, Kazana 1, Maridi.

No specific methods to combat the blackflies were reported, and it was mentioned that the local personal protective measures were insufficient to reduce the biting.

“There is nothing we can do, in the past when we were going to the farms we have to mix some soil with water and apply it to our legs and arms, if you did not do that you will not able to work”. A 30-year-old man, Tarawa Village.

“They said apply diesel to the body but still they (blackflies) bite, they said also wear trousers and long dresses but still they bite”. A 40-year-old woman, Matara Village.

The participants discussed the strategy of slash and clear and they were not convinced it would be an effective intervention in reducing the blackfly population. Some had other options they felt could work in this environment.

“I don’t agree with that (slash and clear), there are holes where these flies live, how can you remove them using panga (machetes)? Those youths who are on training to remove the grass, let them just apply the medicine to the holes where these flies breed and they will finish.” A 42-year-old woman, Kazana 2 Village.

“If we remove the grass from Kazana (the dam) and it became clean, it’s not the solution still, the only solution is the chemicals (larvicidals), and these chemicals need to be supplied to us also so we can kill them also in their hiding places in the latrines”. A 56-year-old man, Matara Village.

“If you just slash and leave it like that, they will still breed here, it need medicine to be applied to it too”. A 43-year-old woman, Gabat Village.

“In 2000 or something some people came here, Chinese or others I don’t know, they sprayed certain medicines in the dam seriously till they got rid of these flies, when they left, the flies returned again, so if we just remove the grass those who remain far will continue to live“. A 35-year-old woman, Matara Village.

### 3.2. OV16 Testing

The overall OV16 seroprevalence in children was 24.3%, but this varied significantly across villages (*p* < 0.001). OV16 antibodies were detected in 19/96 (19.8%) children aged 3–6 years, and in 16/48 (33.3%) children aged 7–9 years (Table 1). 

Kazana 2 (the village closest to the dam) which had the greatest epilepsy burden during previous surveys [8] was also found to have the highest OV16 seroprevalence: 40.0% in children aged 3–6 years, and 66.7% among 7–9 year olds (Figure 3). Regarding ivermectin coverage, 52.8% (47/89) of the tested children aged 5 years and above had taken ivermectin in 2019.

### 3.3. Entomological Investigations

#### 3.3.1. Search for Breeding Sites on the Maridi River

A total of 50 larvae and pupae were collected from one site at the dam spillway out of the 26 sites visited on the Maridi River and its tributaries (see Appendix A). *S. damnosum* larvae and pupae were found at only one site on the Maridi River (Figure 2).

Two sites deserve particular attention:The Maridi River is perennial and originates from the border of the Republic of South Sudan with the Democratic Republic of the Congo (DRC). The river is predominantly covered with papyrus swamp with only a few sites with open water. Downstream the Maridi River is joined by the Mbalala River on the Maridi−Yei road (N: 4°53′10.2″; E: 29°28′6.6″) which is equally dominated by an extensive papyrus swamp with sluggish water flow. This river is the main source of water for Maridi town. The dam has a spillway measuring 132 m across where the water flows down a constructed concrete face falling some 6 m to join the river flowing northwards. This spillway with an estimated water discharge of about 0.4 m^3^/sec was overgrown with vegetation, and numerous larvae and pupae were found attached to the vegetation. In some areas of the spillway some pupae were collected attached to the concrete surface.The second site on the Maridi River was located some 4.5 km downstream from the Maridi dam and 1 km southwest of Maridi State Hospital. This site at Temeregia foot-crossing (N: 4°55′8.5″; E: 29°27′36.9″) was the only site observed to be open with fast flowing water. At this site water flows over rocks covered with a carpet of algae, making it unsuitable for *S. damnosum* breeding. The remaining 24 of the 26 sites visited were all characterized by extensive swamps with poor water flow.

Attempts were not made to further survey the Maridi River from the Rumbek road northwards due to the limited number of days; but this would not have yielded much as the river course is extensively covered with papyrus swamps.

#### 3.3.2. Human Landing Catches of Adult Biting *Simulium* along the Maridi River

The catches conducted during the seven days indicated high biting rates from the three sites. The mean daily biting rates at Maridi dam, Kazana 2 and Matara sites were 202 flies/man/h, 163.6 flies/man/h and 122 flies/man/h, respectively (Kruskal Wallis *p*-value = 0.021) (Table 2). Mean daily biting rate (DBR) was higher at the Maridi dam situated closest to the dam spillway, and lower at the Matara site which is about 3.5 km downstream.

Diurnal fly biting rates were observed to have two peaks: a minor peak from 8:00–9:00 and the second more important peak from 15:00–18:00; while during the remaining hours the biting rates remained relatively low (Figure 4).

#### 3.3.3. Dissection of Flies for Age Group Determination (Parity)

Of the 408 adult female *S. damnosum* caught, 400 were dissected (Table 3). Of these 13.0% were parous, and parity rate ranged from 13.1%–15.7%. Nearly all the parous flies appeared to be young, with opaque malpighian tubules and fat bodies present. In a few cases fresh blood was observed, an indication that they had fed on a human being prior to capture. No fly with retained eggs was observed.

## 4. Discussion

Our study suggests that S. *damnosum* breeding sites appear to be confined to the Maridi dam where favorable conditions to support their breeding were observed. The highest biting rates were also observed close to the dam. In addition to biting rates, OV16 seroprevalence and epilepsy prevalence decreased with increasing distance from the breeding sites. In Kazana 2, the village located closest to the dam and with the highest epilepsy prevalence (11.9%), the prevalence of OV16 antibodies in children aged 3–9 years reached 50%. These data indicate that the Maridi dam is the hotspot for *O. volvulus* transmission in the Maridi area, and suggest a correlation between onchocerciasis transmission and epilepsy prevalence.

One distinct type of larvae was collected from the dam spillway which had pronounced tubernacules and spatulate scales associated with the anthropophilic *S. damnosum*; however, the exact cytospecies remains to be identified. The rest of the sites on the Maridi River surveyed were choked with papyrus, *Cyperus papyrus*. Water outflowing from tracks of papyrus is almost totally de-oxygenated and extremely low in nutrients. Walsh et al. reported a similar situation in central Uganda associated with Lakes Kyoga and Victoria that have sluggish flow and indeed there is no onchocerciasis in central Uganda [19]. One of the sites on the Maridi River with fast flowing water had no breeding of *S. damnosum* which was attributed to the layers of algae coating the rock surface and other substrates. Algae is known to reduce the amount of oxygen and makes it unfavorable for *S. damnosum* breeding. The algae factor was earlier reported to have prevented the re-infestation of the Victoria Nile focus in central Uganda when the vector, *S. damnosum s.s.* was eradicated using DDT in the early 1970s [20].

The two diurnal biting peaks observed at the three study sites reflect the similarity in the microclimate in these sites. In the Sudan, in the Galabat focus in the eastern part of the country, a bimodal daily biting pattern was reported in the mornings and late afternoons coinciding with human activities like farming, grazing of livestock, washing of clothes along the river and collection of water for drinking [21]. Along the Maridi River, as observed, the biting peaks definitely affect community activities. However, the pattern may vary slightly during the rainy season and this will be observed from the monitoring data that will be generated during the course of 2020.

Generally, onchocerciasis-endemic sites like Maridi are prone to also have high epilepsy prevalences [22]. It was also known in the past that living close to rapid flowing rivers could cause blindness and very few people were living in these areas; certain villages close to blackfly breeding sites were even abandoned. A meta-analysis of epilepsy prevalence surveys in West Africa showed that OAE probably existed in West Africa before and during the initial years of vector control but that the epilepsy prevalence decreased gradually with vector control and mass treatment with ivermectin [23]. The low ivermectin coverage among the young children in Maridi indicates that there is an urgent need to step up onchocerciasis elimination actions in the Maridi area, if the epilepsy burden is to be reduced as well.

Several factors may play a role in the development of an OAE epidemic [22]. In 1949 in Mvolo, South Sudan, very high blackfly biting rates were documented but at that time the village was completely deserted; only a police post was present with little or no residential settlements [24]. With the increase in population and poverty, people moved to fertile land close to the river and overlooked the associated health risks of living there. In Kitgum and Pader Districts of northern Uganda, an onchocerciasis-endemic area where there was no ivermectin MDA program initially, the fact that the local Acholi families used to live in close contact with their cows reduced the human blood index (proportion of blood meals taken on humans), thereby reducing the likelihood of developing high microfilarial loads and reducing the risk of developing epilepsy [22]. Moreover cattle, particularly when infected with *Onchocerca ochengi*, confer some protection against *O. volvulus* infection in humans via cross-immunity mechanisms [25]. In 1986, when 300,000 cows were stolen from the Acholi, 1.8 million people were lodged in internally displaced persons (IDP) camps, several located close to rivers with blackfly breeding sites, and the absence of an ivermectin MDA program may have been the main reason why an OAE epidemic appeared in northern Uganda during that period [22]. Upon strengthening onchocerciasis elimination efforts in this area from 2012, the burden of OAE and nodding syndrome was drastically reduced [26].

Our study in Maridi suggests that the construction of the Maridi dam and particularly its repair in the year 2000, together with the interruption of the MDA of ivermectin, may have played a major role in causing the OAE epidemic. High biting rates by blackflies have been reported in the environment of dam spillways and were shown to increase onchocerciasis transmission [27,28,29,30,31]. Six cases of onchocerciasis were diagnosed in expatriates working on a hydroelectric dam project in Taabo (Ivory Coast) between 1977 and 1978 [32]. However, the construction of a dam may also reduce the transmission of onchocerciasis. This was shown with the Merowe dam in the Abu Hamed onchocerciasis focus in northern Sudan [33]. The artificial lake of the dam flooded all the breeding sites in the western region of the focus and no aquatic stages and/or adult blackfly activity were established upstream of the dam which may have contributed to the interruption of onchocerciasis transmission in this area.

## 5. Conclusions

The breeding of *S. damnosum* in Maridi appears to be confined to the Maridi dam spillway. With increasing distance from the dam, there is a progressive decrease in blackfly biting rates, OV16 seroprevalence, and epilepsy prevalence. Historically, blackfly nuisance increased after the construction of the dam in the 1950s, while the burden of epilepsy (including nodding syndrome) reportedly started rising abnormally from the 1990s. It appears that the establishment of the dam favored the breeding of blackflies, which in the absence of optimal onchocerciasis control measures, paved the way for an OAE epidemic. Indeed, the epilepsy prevalence in the Maridi area [8] is much higher than median epilepsy prevalence in sub-Saharan Africa (including several settings non-endemic for onchocerciasis), estimated at 1.4% [34]. In a bid to confirm the role of *O. volvulus* in increasing the local epilepsy burden, onchocerciasis control interventions (biannual treatment with ivermectin, vector control using the slash and clear method [35]) have been implemented in Maridi and their effects on blackfly biting rates, onchocerciasis transmission and epilepsy incidence will be evaluated prospectively [36].

## Figures and Tables

**Figure 1 pathogens-09-00315-f001:**
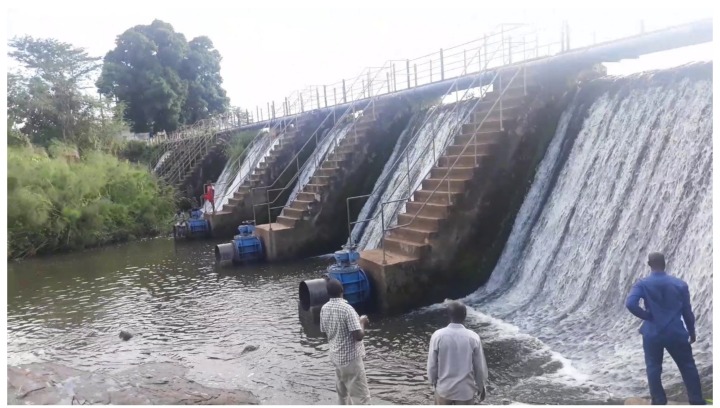
Research team at the Maridi dam, November 2019.

**Figure 2 pathogens-09-00315-f002:**
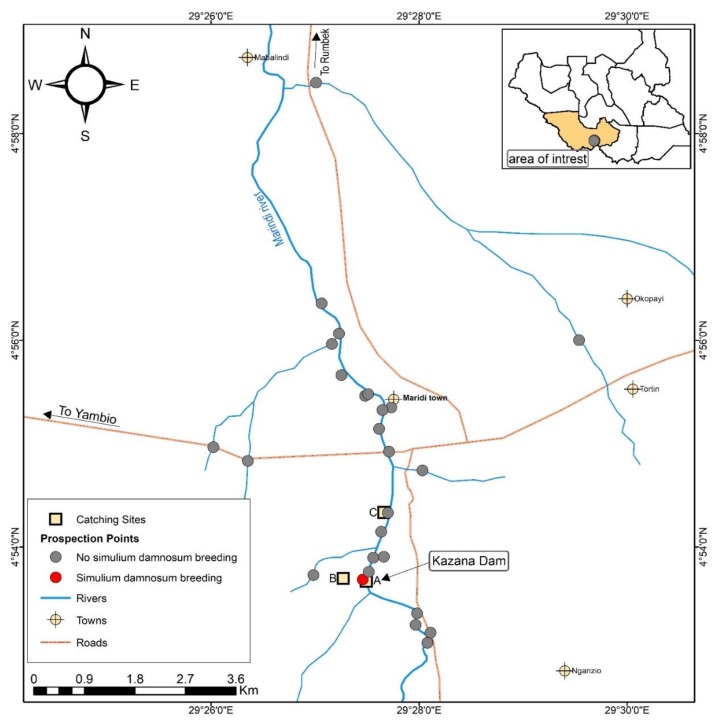
Map showing prospected areas on the Maridi River and associated tributaries in central Maridi County, South Sudan.

**Figure 3 pathogens-09-00315-f003:**
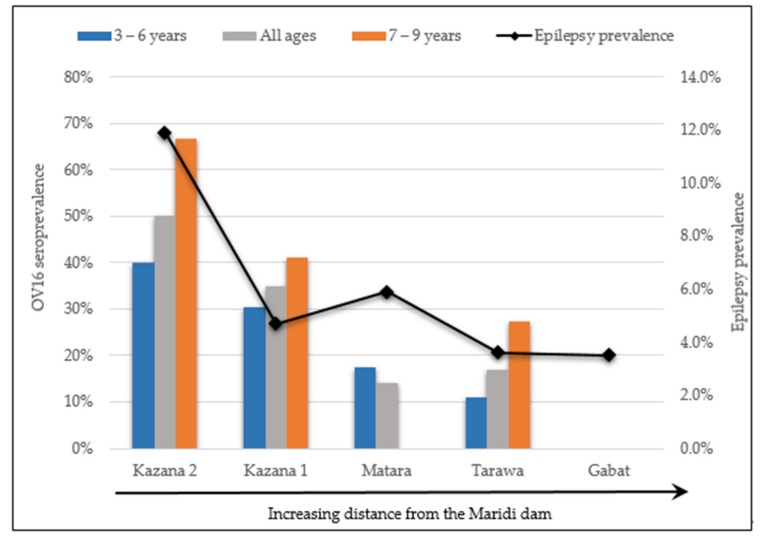
OV16 seroprevalence and epilepsy prevalence in villages of Maridi, South Sudan.

**Figure 4 pathogens-09-00315-f004:**
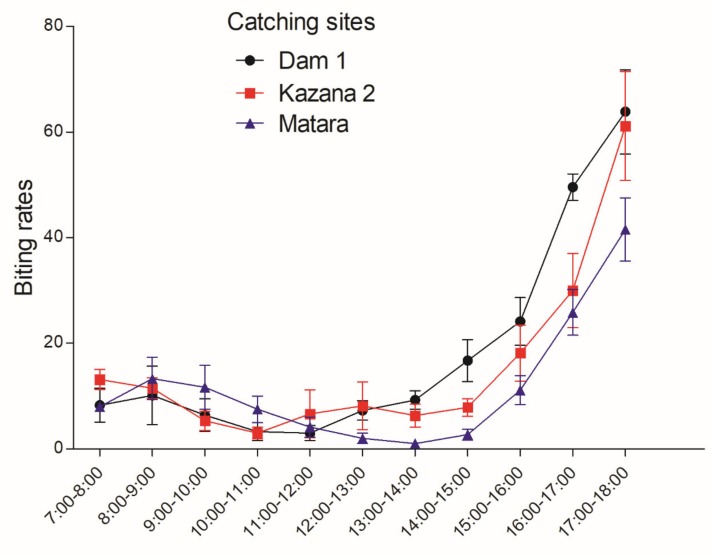
Diurnal biting rate of *S. damnosum* along the Maridi River at the beginning of the dry season, South Sudan. The points indicate mean hourly biting rates over a 7-day period; the error bars represent the standard error of the mean.

**Table 1 pathogens-09-00315-t001:** Age- and sex-specific OV16 seroprevalence in villages in Maridi.

	OV16 Seroprevalence					Comparisons	
	Gabat	Kazana 1	Kazana 2	Matara	Tarawa	Overall	*p*-Value *
**Age-groups**							
3–6 years	0/17 (0%)	7/23 (30.4%)	6/15 (40.0%)	4/23 (17.4%)	2/18 (11.1%)	19/96 (19.8%)	0.074
7–9 years	0/7 (0%)	7/17 (41.2%)	6/9 (66.7%)	0/4 (0%)	3/11 (27.3%)	16/48 (33.3%)	
**Gender**							
Male	0/14 (0%)	10/26 (38.5%)	4/10 (40.0%)	3/11 (27.3%)	2/13 (15.4%)	19/74 (25.7%)	0.694
Female	0/10 (0%)	4/14 (28.6%)	8/14 (57.1%)	1/16 (6.3%)	3/16 (18.8%)	16/70 (22.9%)	

* Chi-squared test on overall OV16 seroprevalence.

**Table 2 pathogens-09-00315-t002:** Human landing catches along the Maridi River during the assessment in November–December 2019.

No	Catching Site		Dates for Catches in November−December 2019						Total	Mean DBR
		29/11	30/11	2/12	3/12	4/12	5/12	6/12	7 days	
1	Maridi dam	228	147	271	219	143	176	230	1414	**202.0**
2	Kazana 2	162	120	84	207	196	193	183	1145	**163.6**
3	Matara	100	72	157	118	93	145	169	854	**122.0**

DBR: Daily Biting Rate (flies/man).

**Table 3 pathogens-09-00315-t003:** Dissection of *S. damnosum* to determine parity rate along the Maridi River.

Catching Site	No. of Flies Caught	No. Dissected	No. Parous	Parity Rate (%)
Maridi dam	143	140	19	13.1
Kazana 2	147	145	15	10.7
Matara	118	115	18	15.7
**Total**	**408**	**400**	**52**	**13.0**

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
