# Peer review of "The Role of the Maridi Dam in Causing an Onchocerciasis-Associated Epilepsy Epidemic in Maridi, South Sudan: An Epidemiological, Sociological, and Entomological Study"

_pathogens, 2020, doi:10.3390/pathogens9040315_

Round 1

Reviewer 1 Report

The authors present an ecological study and suggest an association between onchocerciasis and epilepsy based upon a decreasing prevalence of seropositivity with increasing distance from the source, and identification of breeding sites and high black fly biting rates in the immediate region. They will be assessing incidence of epilepsy in the future following eradication procedures including slash and clear and ivermectin treatment biannually, and while this intervention may provide further evidence to support an epilepsy relationship, a clear causal association will remain elusive without a definitive causal hypothesis.

Do the authors have a hypothesis for the 40 year delay in apparent onset of symptoms? 

Author Response

Response

Our study confirms the many other studies that have recently be published showing the association between onchocerciasis and epilepsy (1-3). Particularly in South Sudan onchocerciasis-associated epilepsy is an major public health problem in onchocerciasis endemic villages (4).

The dam was first build in 1955 but was repaired in early 2000. It is probably the new dam that increased the number of epilepsy cases in the area dramatically. We now specify this in the paper

  1. Colebunders R, Siewe Fodjo JN, Hopkins A, Hotterbeekx A, Lakwo TL, Kalinga A, Logora MY, Basáñez MG. From river blindness to river epilepsy: Implications for onchocerciasis elimination programmes. PLoS Negl Trop Dis. 2019 Jul 18;13(7):e0007407
  2. Gumisiriza N, Mubiru F, Mbonye KM, Hotterbeekx A, Idro R, Makumbi I, Lakwo T, Opar B, Kaducu J, Wamala JF, Siewe Fodjo JN, Colebunders R. Prevalence and incidence of Nodding Syndrome and other forms of epilepsy in onchocerciasis-endemic areas in northern Uganda after the implementation of onchocerciasis control measures. Infect Dis Pov. 2020 Mar 2;9(1):12.
  3. Siewe Fodjo JN, Remme JHF, Preux P-M, Colebunders R. Meta-analysis of epilepsy prevalence in West Africa, and relationship with onchocerciasis endemicity and control. International Health. 2020 Mar 6. pii: ihaa012. doi: 10.1093/inthealth/ihaa012.
  4. Colebunders R, Carter JY, Olore PC, Puok K, Bhattacharyya S, Menon S, Abd-Elfarag G, Ojok M, Ensoy-Musoro C, Lako R, Logora MY. High prevalence of onchocerciasis-associated epilepsy in villages in Maridi County, Republic of South Sudan: a community-based survey. Seizures 2018 Dec;63:93-101.

Reviewer 2 Report

In this study, the authors try to show the effect of the Maridi dam in S. damnosum breeding and causing onchocerciasis-associated epilepsy epidemic in South Sudan. However, overall conclusions are not supported by data and it raises some concerns, 

  1. The main limitation of the study is the lack of strong statistical correlation analysis. The authors should provide a statistical correlation between Ov16 seroprevalence, biting rate, and epilepsy prevalence, or by multivariate analysis.
  2. The authors should elaborate on the method of epilepsy detection. Were those epileptic cases confirmed by any clinician or neurologist? If not authors should mention “suspected cases of epilepsy” rather than “epilepsy”. Studies done on the same geographical area showed an insignificant difference in epilepsy cases (PMID: 30468964). How do the authors explain those findings?
  3.  Although the authors showed an increased biting rate with the proximity of the S. damnosum breeding ground, it will be informative to show the percent of S. damnosum infected with Onchocerca volvulus by field dissection or O-150 PCR method.
  4. Ov16 assay detects the IgG4 titer that takes time to develop and thus not reflect early exposure to O. volvulus. It will be informative to validate the parasitic load by other methods like microscopic examination of skin biopsies (snips) or ELISA assay.
  5. Authors should mention what proportion of patient participated in this study are treated with ivermectin? 

Author Response

In this study, the authors try to show the effect of the Maridi dam in S. damnosum breeding and causing onchocerciasis-associated epilepsy epidemic in South Sudan. However, overall conclusions are not supported by data and it raises some concerns, 

  1. The main limitation of the study is the lack of strong statistical correlation analysis. The authors should provide a statistical correlation between Ov16 seroprevalence, biting rate, and epilepsy prevalence, or by multivariate analysis.

Response

In our conclusion we say that: “With increasing distance from dam, there is a progressive decrease in blackfly biting rates, OV16 seroprevalence, and epilepsy prevalence.”

This statement is based on the following findings:

The OV16 prevalence was statistically higher in the village close to the dam, idem for the blackfly biting rates. Moreover in a previous study we documented that the prevalence of epilepsy was higher in Kazana 2 (11.9%), the village closest to the dam, compared with the other villages.

Our study confirms what has been documented in many other onchocerciasis-endemic regions with high ongoing O. volvulus transmission.

  1. The authors should elaborate on the method of epilepsy detection. Were those epileptic cases confirmed by any clinician or neurologist? If not authors should mention “suspected cases of epilepsy” rather than “epilepsy”. Studies done on the same geographical area showed an insignificant difference in epilepsy cases (PMID: 30468964). How do the authors explain those findings?

Response

In this paper, we do not present data on epilepsy. We only refer to a previous paper which already documented the epidemiology of epilepsy in those same villages (Ref: Colebunders et al, 2018; High prevalence of onchocerciasis-associated epilepsy in villages in Maridi County, Republic of South Sudan: A community-based survey. PMID: 30468964). This paper was written by our research group. The persons with epilepsy in Maridi were all diagnosed by clinical officers trained in the diagnosis of epilepsy and by two medical doctors who also supervised the clinical officers.

We do not understand what the reviewer means with “insignificant difference in epilepsy cases.” The paper PMID: 30468964 clearly indicates that villages closer to the dam have higher epilepsy prevalence, and that increased duration of stay in the onchocerciasis-endemic villages also resulted in more epilepsy cases.

We wrote another paper about epilepsy in the Maridi area that contains more clinical details about the epilepsy cases: (Ref: Colebunders et al, 2018. Clinical characteristics of onchocerciasis-associated epilepsy in villages in Maridi County, Republic of South Sudan. Seizure; 62:108-115. doi: 10.1016/j.seizure.2018.10.004.) This paper is also included in the references. We therefore used the data from these two initial studies when conducting the present study. Given that epilepsy is a chronic disease and that the population changes are minimal in these villages, we do not expect the epilepsy situation in 2019 to differ much from our previous findings of 2018.

  1. Although the authors showed an increased biting rate with the proximity of the  damnosum breeding ground, it will be informative to show the percent of S. damnosum infected with Onchocerca volvulus by field dissection or O-150 PCR method.

Response

We agree that it would be interesting to perform O-150 PCR testing on the flies. However given the COVID-19 pandemic this is currently impossible to organize. We plan to do this once we have one year follow-up data of the slash and clear intervention

  1. Ov16 assay detects the IgG4 titer that takes time to develop and thus not reflect early exposure to  volvulus.It will be informative to validate the parasitic load by other methods like microscopic examination of skin biopsies (snips) or ELISA assay.

Response

We agree a positive OV16 test only shows that the person has acquired an O. volvulus infection in the past. However if the tested children are less than < 10 years old, this means it was in the recent past, suggesting that there is high ongoing O. volvulus transmission in the area. This is indeed the case, as we also have done skin snip testing in persons with epilepsy and controls in the area. Results showed very high infection rates of skin snip; positivity (up to 50% of controls were infected) and high microfilarial loads particularly in persons with epilepsy living in the area close to the Maridi dam

A paper about these data is in press

(Ref: Abd-Elfarag G, Carter JY, Raimon S, Sebit W, Suliman A, Siewe Fodjo JN, Claver Olore P, Biel KP Morrish Ojok, Logora MY, Colebunders R. Persons with onchocerciasis-associated epilepsy with nodding seizures have a more severe form of epilepsy, with more cognitive impairment and higher levels of Onchocerca volvulus infection. Epileptic Disorders 2020; in press.

  1. Authors should mention what proportion of patient participated in this study are treated with ivermectin?

Response

Based on the previous studies in Maridi, ivermectin coverage was 40.8% in 2018 (Ref: Colebunders et al, 2018; High prevalence of onchocerciasis-associated epilepsy in villages in Maridi County, Republic of South Sudan: A community-based survey. PMID: 30468964). In the present study, we investigated ivermectin use only in participants who were tested for Ov16 antibodies; we found a coverage of 52.8% (47/89) in the 5-9 year old children. This has now been mentioned in the manuscript.

Round 2

Reviewer 2 Report

Authors are suggested to provide the total number of children aged 3−6 years and 7−9 years used for OV16 testing in each village in a tabulated format. Does the OV16 seroprevalence depend on the age group or gender?

Please mention in the manuscript what is the prevalence of epilepsy in NON-onchocerciasis endemic regions in South Sudan.

In Figure 2, please place the data points (black) representing epilepsy prevalence on the gray bar that representing the "All ages" average NOT on the orange bar (7−9 years age group).

In the legend of figure2 and figure 4 mention which the statistical test was used for that figure.

In Figure4, the authors should show the standard error for each point. Separate the total daily biting rate (last point) from the graph. Plot a separate diagram showing the total biting rate (with SEM) vs the distance from Maridi dam (different study sites). Did the authors find any significant difference in the biting rate depend on the age group or gender?

Author Response

Response to reviewer

Authors are suggested to provide the total number of children aged 3−6 years and 7−9 years used for OV16 testing in each village in a tabulated format. Does the OV16 seroprevalence depend on the age group or gender?

Response

We have now introduced a new Table 1 which details age- and sex-specific OV16 seroprevalence in each village.

Please mention in the manuscript what is the prevalence of epilepsy in NON-onchocerciasis endemic regions in South Sudan.

Response

The prevalence of epilepsy in non-onchocerciasis endemic regions in South Sudan is not known but the median epilepsy prevalence in sub Saharan Africa is 1.4%. We now include this information in the paper.

In Figure 2, please place the data points (black) representing epilepsy prevalence on the gray bar that representing the "All ages" average NOT on the orange bar (7−9 years age group).

Response

Done

In the legend of figure2 and figure 4 mention which the statistical test was used for that figure.

Response

We do not understand this question, given that we did not present any statistical analysis on those figures. The details of the statistical tests used in this study are given in the methods, section “Data analysis”.

In Figure4, the authors should show the standard error for each point. Separate the total daily biting rate (last point) from the graph. Plot a separate diagram showing the total biting rate (with SEM) vs the distance from Maridi dam (different study sites).

Response

We have now added SEM and removed the last point from the rest of the graph. However, we cannot present data by village because we had only three catching sites (represented by the three lines) and the distances of each point from the dam are well specified in the manuscript.

Did the authors find any significant difference in the biting rate depend on the age group or gender?

Response

To determine biting rates, we used human volunteers (young males in their twenties). They all had a similar age and gender. Therefore we cannot respond to your question. However, the reviewer may be interested in various mathematical models that have generated age and sex-dependent exposure functions for onchocerciasis (Ref: Hamley JID, et al. Structural Uncertainty in Onchocerciasis Transmission Models Influences the Estimation of Elimination Thresholds and Selection of Age Groups for Seromonitoring. The Journal of Infectious Diseases. 2020;jiz674).

Round 3

Reviewer 2 Report

English language and style are fine/minor spell check required